# Stability Analysis of Singly Diagonally Implicit Block Backward Differentiation Formulas for Stiff Ordinary Differential Equations

**Saufianim Jana Aksah** [1], **Zarina Bibi Ibrahim** [2,*] **and Iskandar Shah Mohd Zawawi** [3]

1   Department of Mathematics, Faculty of Science, Universiti Putra Malaysia,
    43400 UPM Serdang, Selangor, Malaysia; saufianim2412@yahoo.com
2   Institute for Mathematical Research, Universiti Putra Malaysia, 43400 UPM Serdang, Selangor, Malaysia
3   Faculty of Computer and Mathematical Sciences, Universiti Teknologi MARA, Seremban Campus,
    70300 Seremban, Negeri Sembilan, Malaysia; iskandarshah@uitm.edu.my
*   Correspondence: zarinabb@upm.edu.my; Tel.: +603-89466861

**Abstract:** In this research, a singly diagonally implicit block backward differentiation formulas (SDIBBDF) for solving stiff ordinary differential equations (ODEs) is proposed. The formula reduced a fully implicit method to lower triangular matrix with equal diagonal elements which will results in only one evaluation of the Jacobian and one LU decomposition for each time step. For the SDIBBDF method to have practical significance in solving stiff problems, its stability region must at least cover almost the whole of the negative half plane. Step size restriction of the proposed method have to be considered in order to ensure stability of the method in computing numerical results. Efficiency of the SDIBBDF method in solving stiff ODEs is justified when it managed to outperform the existing methods for both accuracy and computational time.

**Keywords:** singly diagonally implicit; block multistep method; A-stable; step-size restriction; stiff ODEs

---

## 1. Introduction

Many problems in engineering, physical and social sciences are reduced to quantifiable form through mathematical modelling involving systems of ordinary differential equations (ODEs). These problems sometimes exhibit a phenomenon known as stiffness. It associates with components that are decaying at widely differing rates. For this article, our main concern is the linear system of first order stiff ODEs of the form

$$y'(x) = f(x, y), \quad y(a) = \mu, \quad x \in [a, b], \tag{1}$$

where $y^T = (y_1(x), y_2(x), \ldots, y_d(x))$, $f^T = (f_1(x), f_2(x), \ldots, f_d(x))$ and $\mu^T = (\mu_1(x), \mu_2(x), \ldots, \mu_d(x))$. Equation (1) is said to be linear if $f(x, y) = A(x)y + \Phi(x)$, where $A(x)$ is a constant $d \times d$ matrix and $\Phi(x)$ is an $d$-dimensional vector.

There are various definitions of stiffness that exist in the literature. However, we consider the one given by [1] which stated that the linear system (1) is said to be stiff if

1.   $Re(\lambda_i) < 0, i = 1, 2, \ldots, d$ and

2.   $\max_i |Re(\lambda_i)| >> \min_i |Re(\lambda_i)|$, where $\lambda_i$ are the eigenvalues of $A$ and the ratio $S = \frac{\max_i |Re(\lambda_i)|}{\min_i |Re(\lambda_i)|}$ is called the stiffness ratio.

Most realistic stiff systems do not have analytical solutions, hence a numerical approach have to be used [2]. An exact solution of (1) can be approximated by using either the one-step method or the linear multistep method (LMM). Euler's is the simplest form of one-step method whereas the Runge-Kutta (RK) method is the most famous family under this class. These methods use the solution of current point, $y_n$, as initial value to compute solution at the next point, $y_{n+1}$.

On the contrary, LMM uses information from the previous points to calculate the next values. Adams and BDF method are examples of families under the LMM. A classical approach for numerical methods in finding an approximation of $y_{n+1}$ at the point $x_{n+1}$ is computed one step at a time.

In order to accelerate the computational process, block method is introduced with the idea of producing simultaneously $k$-blocks where each block contains $r$-point approximation, $y_{n+1}, y_{n+2}, \ldots, y_{n+r}$, at each iteration of the algorithm. For the multistep block method, points from the previous block calculated are used in generating approximation solutions for the new block. The block method is first proposed by [3] which is later extended by several researchers such as [4–6]. A modified block by Adams method for higher order ODEs had been discussed by [7] followed by [8] for solving nonstiff higher order ODEs.

As for stiff ODEs, fixed coefficients BBDF introduced by [9] had proved to optimize performance in both accuracy and computational time when solving stiff problems. Not only that, but the method also managed to outperform the non-block variable step variable order BDF (VSVOBDF) method by [10] (as cited in [9]). Numbers of improvement had been made to the original BBDF method, one of them is the diagonally implicit 2-point BBDF (DI2BBDF) method by [11] that gives better accuracy for high dimension problem of ODEs than the BBDF method. The method is derived based on the motivation by [12] which defined the diagonally implicit as the method with its coefficients of the upper-diagonal (read: lower triangular) entries are zero.

Over the years, conventional RK methods evolutionized to form the best solver for ODEs. The commonly improved RK methods involved an implementation of singly diagonally implicit properties which had proved to produce better computational time when compared with the existing methods. Among the initial works on the singly diagonally implicit RK (SDIRK) method is [13], which defined a singly diagonally implicit as a method with equal diagonal elements, $\alpha_{ii} = \gamma$, as shown in the Butcher tableau below.

$$
\begin{array}{c|cccc}
c_1 & \gamma & 0 & \ldots & 0 \\
c_2 & \alpha_{21} & \gamma & \ldots & 0 \\
\vdots & \vdots & \vdots & & \vdots \\
c_n & \alpha_{n1} & \alpha_{n2} & \ldots & \gamma \\
\hline
 & b_1 & b_2 & \ldots & b_n
\end{array}
$$

The proposed problem of stiff ODEs is also considered by [14] through Newton-type iterations that solve for linear systems at each stage with coefficient matrix of the form $I - h\alpha_{ii}\frac{\partial f}{\partial y}$. Based on [15], singly implicit RK is a transformed method whose RK matrix has just one real $s$-fold eigenvalue. If we have $\alpha_{ii} = \gamma$, then the class of diagonally implicit RK (DIRK) is called SDIRK methods [16].

The definition of SDIRK by Norsett is further discussed by [17–19] in their research. As summarized by [20], when having the lower triangular matrix with equal diagonal elements, the stored LU-factorization of a single such matrix can be used repeatedly. Hence, only one evaluation of the Jacobian and one LU decomposition will be needed for each time step. By having these properties, degree of implicitness can also be reduced as it involved less computational process which will results in less execution time [21].

Therefore, developing the SDIBBDF method involves the hybrid process of implementing qualities from SDIRK method to block multistep method. As we know, both methods are of different families hence, our main concern is the compatibility of the derived method to solve for stiff ODEs. Derivation of the SDIBBDF method is shown in the next section.

## 2. Research Methodology

### 2.1. Derivation of 2-Point SDIBBDF Method

Our main objective is to develop a method that capable of computing two solutions, $y_{n+1}$ and $y_{n+2}$, simultaneously in less expensive environment with accurate approximation to the exact solution of stiff ODEs. The idea is illustrated as shown in Figure 1.

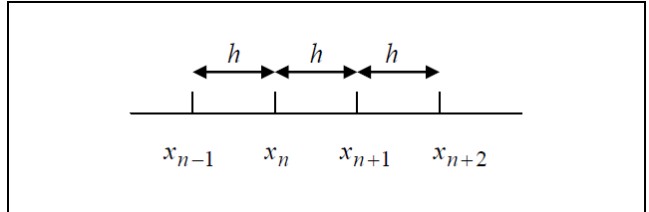

**Figure 1.** 2-point SDIBBDF method of constant step size.

Points $x_{n-1}$ and $x_n$ are the back values used to evaluate solutions of future points, $x_{n+1}$ and $x_{n+2}$, with constant step size. Hence, we proposed the 2-point SDIBBDF method of order 2 with

$$\sum_{j=0}^{k+s-2} \alpha_{s,j} y_{n+j} = h\beta_{s,k+s-2} f_{n+s} \tag{2}$$

where $k = 2$, and $s = 1,2$ for points $y_{n+1}$ and $y_{n+2}$ respectively with $\alpha_{ii} = \gamma$, as referred in [13]. We consider the linear difference operator for block multistep method (2) as

$$L_s(y(x) : h) = \sum_{j=0}^{k+s-2} (\alpha_{s,j} y(x + jh) - h\beta_{s,k+s-2} y(x + sh). \tag{3}$$

Equation (3) is expanded by using the Taylor series, and the terms of derivative $y$ are collected to produce

$$L_s(y(x) : h) = C_0 y(x) + C_1 y'(x) + \dots + C_q y^q(x) + \dots. \tag{4}$$

The constant $C_q$ is defined as

$$C_0 = \sum_{j=0}^{k+s-2} \alpha_j,$$

$$C_1 = \sum_{j=0}^{k+s-2} j\alpha_j - \beta_j,$$

$$\vdots$$

$$C_q = \left[ \frac{1}{q!} \sum_{j=0}^{k+s-2} j^q \alpha_j - \frac{1}{(q-1)!} \sum_{j=2}^{k+s-2} j^{q-1} \beta_j \right], \quad q = 2,3,\dots$$
$$\tag{5}$$

Since the SDIBBDF method proposed is of order 2, $k = 3$ and $q = 2$ are substituted into Equation (4) to get

$$C_0 = \begin{bmatrix} \alpha_{1,-1} & \alpha_{1,0} & \gamma & 0 \\ \alpha_{2,-1} & \alpha_{2,0} & \alpha_{2,1} & \gamma \end{bmatrix} = 0,$$

$$C_1 = \begin{bmatrix} \alpha_{1,0} & 2\gamma & 0 & -\beta_{1,2} & 0 \\ \alpha_{2,0} & 2\alpha_{2,1} & 3\gamma & 0 & -\beta_{2,3} \end{bmatrix} = 0, \tag{6}$$

$$C_2 = \begin{bmatrix} \frac{1}{2}\alpha_{1,0} & 2\gamma & 0 & -2\beta_{1,2} & 0 \\ \frac{1}{2}\alpha_{2,0} & 2\alpha_{2,1} & \frac{9}{2}\gamma & 0 & -3\beta_{2,3} \end{bmatrix} = 0,$$

By using MAPLE, Equation (6) is solved simultaneously, and the coefficients obtained are substituted into Equation (2). The formula derived is rewritten in matrices form as shown below:

$$
\begin{bmatrix} \dfrac{1}{2} & -2 \\ 0 & \dfrac{1}{2} \end{bmatrix} \begin{bmatrix} y_{n-1} \\ y_n \end{bmatrix} + \begin{bmatrix} \dfrac{3}{2} & 0 \\ -2 & \dfrac{3}{2} \end{bmatrix} \begin{bmatrix} y_{n+1} \\ y_{n+2} \end{bmatrix} = h \begin{bmatrix} \beta_{1,2} & 0 \\ 0 & \beta_{2,3} \end{bmatrix} \begin{bmatrix} f_{n+1} \\ f_{n+2} \end{bmatrix}.
\tag{7}
$$

Therefore, the general formula of 2-point SDIBBDF method of order 2 is as follows:

$$
\begin{aligned}
y_{n+1} &= \frac{1}{3}y_{n-1} + \frac{4}{3}y_n + \frac{2}{3}hf_{n+1}, \\
y_{n+2} &= \frac{1}{3}y_n + \frac{4}{3}y_{n+1} + \frac{2}{3}hf_{n+2}.
\end{aligned}
\tag{8}
$$

In the next subsection, we will discuss on stability of the derived SDIBBDF method.

### 2.2. Stability Analysis

One of the practical characteristics for a method to be useful is that it must have a region of absolute stability.

**Definition 1.** *The LMM in Equation (1) is said to be absolutely stable in a region R for a given H if and only if for that H, all the roots, $r_s = r_s(H)$ of the stability polynomial of the linear k-step method, $\pi(r, H) = \rho(r) - H\phi(r)$, satisfy $|r_s| < 1$, $s = 1, 2, \ldots, k$ where $H = h\lambda$ and $\rho(r)$ and $\phi(r)$ are the first and second characteristic polynomials respectively. Otherwise the method is said to be absolutely unstable.*

In order to analyse the stability properties of the proposed method, a stability graph of the SDIBBDF method has to be constructed. First, characteristic polynomial of the method is determined by referring to the following statement.

**Definition 2.** *The characteristic polynomial of LMM in Equation (1) assumes*

$$
\pi(r, h\lambda) = \rho(r) - h\lambda\phi(r) = 0,
$$

*where $H = h\lambda$ and $\lambda = \dfrac{\partial f}{\partial y}$ is complex.*

By applying the Dahlquist test equation of

$$
\begin{aligned}
y' &= f(x, y) \\
&= \lambda y
\end{aligned}
\tag{9}
$$

to Equation (7), we get

$$
\begin{bmatrix} 1 - \dfrac{2}{3}H & 0 \\ -\dfrac{4}{3} & 1 - \dfrac{2}{3}H \end{bmatrix} \begin{bmatrix} y_{n+1} \\ y_{n+2} \end{bmatrix} = \begin{bmatrix} -\dfrac{1}{3} & \dfrac{4}{3} \\ 0 & -\dfrac{1}{3} \end{bmatrix} \begin{bmatrix} y_{n-1} \\ y_n \end{bmatrix}
\tag{10}
$$

which is equivalent to

$$
AY_m = BY_{m-1},
$$

where $A$ and $B$ are properly chosen $r \times r$ matrix coefficients and $m$ represents the block number.

We consider $\rho(t) = det(At - B)$ as the first characteristic polynomial of the SDIBBDF method. By substituting Equation (10) into $\rho(t)$,

$$\rho(t) = det \begin{bmatrix} \left(1 - \frac{2}{3}H\right)t + \frac{1}{3} & -\frac{4}{3} \\ -\frac{4}{3}t & \left(1 - \frac{2}{3}H\right)t + \frac{1}{3} \end{bmatrix}. \tag{11}$$

Based on Definition 2, when we solve for $\rho(t) = 0$, stability polynomial, $R(H)$, of Equation (8) is obtained.

$$R(H) = \frac{4}{9}t^2H^2 - \frac{4}{3}t^2H + t^2 - \frac{4}{9}tH - \frac{10}{9}t + \frac{1}{9}. \tag{12}$$

In order to determine zero stability of the SDIBBDF method, we refer to the following definition.

**Definition 3.** *The LMM in (1) is said to be zero-stable if no root of the first characteristic polynomial, $\rho(\zeta)$, has modulus greater than one, and if every root with modulus one is simple.*

Solving $R(H) = 0$ yields the following roots:

$$t = 0, 1. \tag{13}$$

Therefore, Equation (13) proved that the SDIBBDF method is zero stable.

To plot $R(H)$, we let for each value of $H$, $R$ is a complex number. Boundary of the stability region is the set of all $H$ such that $R(H)$ is on the unit circle of

$$R(H) = e^{i\theta}, \tag{14}$$

for some $\theta \in [0, 2\pi]$. Equation (14) is expanded for various value of $\theta$ in steps of $\frac{2\pi}{n}$ from 0 to $2\pi$.

Next, the points are plotted by using MAPLE as shown in Figure 2 while Figure 3 represents the output which is the stability graph of 2-point SDIBBDF method.

```
>  restart :
>  R := t² - 4/3 t² H - 10/9 t + 4/9 t² H² - 4/9 t H + 1/9 :
>  % = 0 :
>  subs( t = exp(theta·I), % ) :
>  R1 := solve( %, H );
```

$$R1 := \frac{\frac{3}{2}e^{I\theta} + \frac{1}{2} + 2e^{\frac{1}{2}I\theta}}{e^{I\theta}}, \quad \frac{\frac{3}{2}e^{I\theta} + \frac{1}{2} - 2e^{\frac{1}{2}I\theta}}{e^{I\theta}}$$

```
>  p := 
```
$$\frac{\frac{3}{2}e^{I\theta} + \frac{1}{2} + 2e^{\frac{1}{2}I\theta}}{e^{I\theta}} :$$

```
>  q := 
```
$$\frac{\frac{3}{2}e^{I\theta} + \frac{1}{2} - 2e^{\frac{1}{2}I\theta}}{e^{I\theta}} :$$

```
>  with( plots ) :
>  complexplot( [ p, q ], theta = 0 ..2·Pi, numpoints = 5000,
         colour = red );
```

**Figure 2.** MAPLE codes to construct stability graph of 2-point SDIBBDF method.

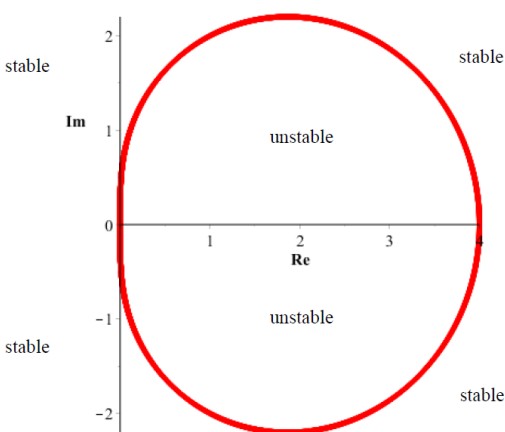

**Figure 3.** Stability graph of 2-point SDIBBDF method.

By referring to [22] a method is suitable for solving stiff problems because of its *A*-stability property as reviewed in the following statement.

**Definition 4.** *A numerical method is said to be A-stable if its region of absolute stability contains the whole left-hand half-plane, $Re(h\lambda) < 0$.*

As observed in Figure 3, the area inside the closed region is unstable whereas the stable part lies outside the region. Coincide with Definition 4, we can conclude that the 2-point SDIBBDF method of order 2 is *A*-stable.

Suppose that the following definition is applied.

**Definition 5.** *The LMM in Equation (1) is said to have region of absolute stability, $\Re_A$, where $\Re_A$ is a region of the complex $\hat{h}$-plane, if it is absolutely stable for all $\hat{h} \in \Re_A$. The intersection with real axis is called the interval of absolute stability.*

For SDIBDDF method, its interval of absolute stability is $4 < H < 0$. Comparison in terms of stability region is made between the proposed method, fully implicit BBDF by [9] and DIBBDF method by [11] as shown in Figure 4.

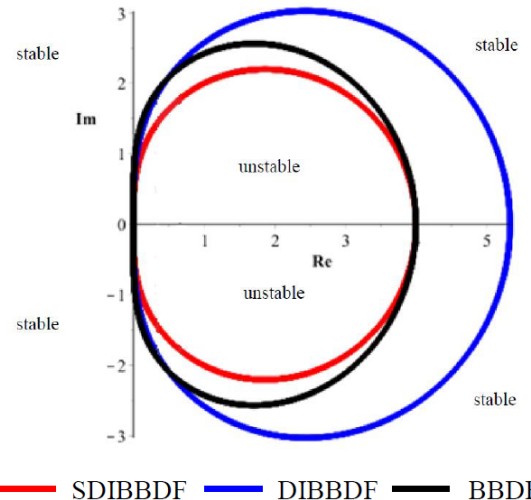

**Figure 4.** Comparison of stability graph between SDIBBDF, BBDF and DIBBDF method.

Analysis on Figure 4 concludes that the unstable region of the proposed method is smaller when compared with the other methods. Interval of the unstable area for each method is presented in Table 1.

**Table 1.** Interval of unstable area on real and imaginary axis.

| METHOD | Re | Im |
|--------|-----|-----|
| SDIBBDF | $[0, 3.99]$ | $[-2.20, 2.19]$ |
| DIBBDF | $[0, 5.33]$ | $[-3.03, 3.01]$ |
| BBDF | $[0, 3.99]$ | $[-2.57, 2.55]$ |

From the table shown, it is noted that the unstable area of DIBBDF method is wider followed by the BBDF and SDIBBDF method. This proved that the SDIBBDF method has wider stability area than the comparing methods.

*2.3. Step Size Restriction*

As stated in Definition 1, an LMM is said to be absolutely stable when $|R(H)| < 1$, otherwise it is unstable. There are two parameters involved, $h$ and $\lambda$, but it is only their product, $H$, that needs to be taken into accounts.

By solving $|R(H)| < 1$, we found that the SDIBBDF method is stable everywhere except when $H \in [0, 4]$. However, $h$ must lies within a certain range in order for the SDIBBDF method to be stable. By substituting endpoint of the interval into characteristic polynomial in Equation (12), we obtain

$$\varepsilon = 7.075600000H^2 - 23.00013333H + 11.59787778. \tag{15}$$

Next, Equation (15) is presented in the form of $|\varepsilon| < 1$ which equal to

$$|7.075600000H^2 - 23.00013333H + 11.59787778| < 1. \tag{16}$$

Solving Equation (16) yields

$$|H| < 0.6240609349.$$

Please note that $H = h\lambda$,

$$h < \left| \frac{0.6240609349}{\lambda} \right|. \tag{17}$$

Thus, Equation (17) is the step size restriction of SDIBBDF method. By taking an example of a stiff ODEs with eigenvalue, $\lambda = -100$,

$$h < \left| \frac{0.6240609349}{-100} \right|,$$

the suitable step size for solving the problem must be

$$h < 0.006240609349.$$

If we take $h > 0.006240609349$, the method will be unstable therefore, numerical solutions with large maximum error is produced.

**3. Numerical Results**

In the previous section, we claimed that the proposed method is $A$-stable hence, suitable for solving stiff problems. To validate this finding, we tested the SDIBBDF method to solve for single and system of stiff ODEs. Efficiency of the method is measured in terms of its accuracy and execution time.

Accuracy is analysed based on the maximum error,

$$MAXE = \max_{1 \leq i \leq T} \left( \max_{1 \leq i \leq N} (error_i)_t \right), \tag{18}$$

where $T$ is the total number of steps and $N$ is the number of equations. By considering the error as

$$(error_i)_t = |(y_i)_t - (y(x_i))_t|,$$

with $y_i$ and $y(x_i)$ are the approximate and exact solutions of Equation (8) respectively.
The following notations are used

| | | |
|---|---|---|
| SDIBBDF | : | Singly diagonally implicit BBDF method |
| DIBBDF | : | Diagonally implicit BBDF method by [23] |
| BBDF | : | BBDF method by [24] |
| ode15s | : | VSVO solver based on the numerical differentiation formulas (NDFs) |
| ode23s | : | modified Rosenbrock formula of order 2 |
| $h$ | : | step size |
| MAXE | : | maximum error |
| TIME | : | computational time |

To verify performance of the methods, the following test problems of stiff ODEs are solved.

**Test Problem 1**

$$y' = -20y + 20 \sin x + \cos x, \qquad\qquad y(0) = 1, \quad 0 \le x \le 2$$

Exact solution:

$$y(x) = \sin x + e^{-20x},$$

Eigenvalue: $\lambda = -20$,
Source: [25].

**Test Problem 2**

$$y' = 100(\sin x - y), \qquad\qquad y(0) = 0, \quad 0 \le x \le 3$$

Exact solution:

$$y(x) = \frac{\sin x - 0.01 \cos x + 0.01 e^{-100x}}{1.0001},$$

Eigenvalue: $\lambda = -100$,
Source: [22].

**Test Problem 3**

$$y'_1 = 32y_1 + 66y_2 + \frac{2}{3}x + \frac{2}{3}, \qquad\qquad y_1(0) = \frac{1}{3}, \quad 0 \le x \le 1,$$

$$y'_2 = -66y_1 - 133y_2 - \frac{1}{3}x - \frac{1}{3}, \qquad\qquad y_1(0) = \frac{1}{3},$$

Exact solution:

$$y_1(x) = \frac{2}{3}x + \frac{2}{3}e^{-x} - \frac{1}{3}e^{-100x},$$

$$y_2(x) = -\frac{1}{3}x - \frac{1}{3}e^{-x} + \frac{2}{3}e^{-100x},$$

Eigenvalues: $\lambda = -100, -1$,

Source: [25].

**Test Problem 4**

$$y_1' = -y_1 + 95y_2, \qquad\qquad y_1(0) = 1, \quad 0 \le x \le 10,$$
$$y_2' = -y_1 - 97y_2, \qquad\qquad y_2(0) = 1,$$

Exact solution:

$$y_1(x) = \frac{1}{47}\left(95e^{-2x} - 48e^{-96x}\right),$$

$$y_2(x) = \frac{1}{47}\left(48e^{-96x} - e^{-2x}\right),$$

Eigenvalues: $\lambda = -1000, -1$,
Source: [26].

**Test Problem 5**

$$y_1' = -21y_1 + 19y_2 - 20y_3, \qquad\qquad y_1(0) = 1, \quad 0 \le x \le 10,$$
$$y_2' = 19y_1 - 21y_2 + 20y_3, \qquad\qquad y_2(0) = 0,$$
$$y_3' = 40y_1 - 40y_2 - 40y_3, \qquad\qquad y_3(0) = -1,$$

Exact solution:

$$y_1(x) = 0.5\left[e^{-2x} + e^{-40x}\left(\cos 40x + \sin 40x\right)\right],$$
$$y_2(x) = 0.5\left[e^{-2x} - e^{-40x}\left(\cos 40x + \sin 40x\right)\right],$$
$$y_3(x) = 2e^{-40x}\left(-\frac{1}{2}\cos 40x + \frac{1}{2}\sin 40x\right),$$

Eigenvalues: $\lambda = -40 - 40i, -40 + 40i, -2$,
Source: [1].

Tables 2–6 represent the numerical approximations of SDIBBDF, DIBBDF, BBDF, ode15s and ode23s for stiff ODEs with various step size. The results are illustrated in the form of efficiency curves as shown in Figures 5–9. Figures 5a–9a indicate performance of the methods based on step sizes tested. Meanwhile, Figures 5b–9b give visual representations on efficiency of the methods in terms of computational time to approximate solutions.

**Table 2.** Numerical results for test problem 1.

| h | Method | Maxe | Time |
|---|---|---|---|
| $10^{-2}$ | SDIBBDF | $4.17749 \times 10^{-2}$ | $1.89443 \times 10^{-6}$ |
| | DIBBDF | $9.19710 \times 10^{-2}$ | $5.74779 \times 10^{-5}$ |
| | BBDF | $7.82684 \times 10^{-2}$ | $6.11443 \times 10^{-5}$ |
| | ode15s | $8.36909 \times 10^{-3}$ | - |
| | ode23s | $4.07991 \times 10^{-3}$ | - |
| $10^{-4}$ | SDIBBDF | $4.94771 \times 10^{-6}$ | $2.10183 \times 10^{-5}$ |
| | DIBBDF | $1.46293 \times 10^{-3}$ | $1.67354 \times 10^{-3}$ |
| | BBDF | $1.46435 \times 10^{-3}$ | $3.76579 \times 10^{-3}$ |
| | ode15s | $1.66322 \times 10^{-4}$ | - |
| | ode23s | $1.83868 \times 10^{-4}$ | - |

**Table 2.** *Cont.*

| h | Method | Maxe | Time |
|---|---|---|---|
| $10^{-6}$ | SDIBBDF | $4.99893 \times 10^{-10}$ | $1.36298 \times 10^{-3}$ |
| | DIBBDF | $1.47125 \times 10^{-5}$ | $9.98032 \times 10^{-2}$ |
| | BBDF | $1.47126 \times 10^{-5}$ | $1.07378 \times 10^{-1}$ |
| | ode15s | $2.75074 \times 10^{-6}$ | - |
| | ode23s | $1.25088 \times 10^{-5}$ | - |
| $10^{-8}$ | SDIBBDF | $4.97015 \times 10^{-10}$ | $1.53342 \times 10^{-1}$ |
| | DIBBDF | $1.47131 \times 10^{-7}$ | $7.19868 \times 10^{0}$ |
| | BBDF | $1.47012 \times 10^{-7}$ | $7.57031 \times 10^{0}$ |
| | ode15s | $2.75074 \times 10^{-6}$ | - |
| | ode23s | $1.25548 \times 10^{-5}$ | - |

**Table 3.** Numerical results for test problem 2.

| h | Method | Maxe | Time |
|---|---|---|---|
| $10^{-2}$ | SDIBBDF | $5.50135 \times 10^{-3}$ | $1.17163 \times 10^{-7}$ |
| | DIBBDF | $9.11949 \times 10^{0}$ | $7.40632 \times 10^{-5}$ |
| | BBDF | $7.32490 \times 10^{-4}$ | $8.05945 \times 10^{-5}$ |
| | ode15s | $1.21111 \times 10^{-4}$ | - |
| | ode23s | $1.70139 \times 10^{-3}$ | - |
| $10^{-4}$ | SDIBBDF | $1.20673 \times 10^{-6}$ | $3.11372 \times 10^{-6}$ |
| | DIBBDF | $7.14971 \times 10^{-5}$ | $3.70624 \times 10^{-3}$ |
| | BBDF | $7.18301 \times 10^{-5}$ | $6.91835 \times 10^{-3}$ |
| | ode15s | $3.20516 \times 10^{-6}$ | - |
| | ode23s | $3.75006 \times 10^{-5}$ | - |
| $10^{-6}$ | SDIBBDF | $1.24891 \times 10^{-10}$ | $1.59917 \times 10^{-4}$ |
| | DIBBDF | $7.35527 \times 10^{-7}$ | $2.47110 \times 10^{-1}$ |
| | BBDF | $7.35563 \times 10^{-7}$ | $7.47109 \times 10^{-1}$ |
| | ode15s | $1.66215 \times 10^{-6}$ | - |
| | ode23s | $2.61572 \times 10^{-6}$ | - |
| $10^{-8}$ | SDIBBDF | $1.23007 \times 10^{-10}$ | $2.62004 \times 10^{-1}$ |
| | DIBBDF | $7.35736 \times 10^{-9}$ | $3.64344 \times 10^{0}$ |
| | BBDF | $7.35729 \times 10^{-9}$ | $4.69932 \times 10^{0}$ |
| | ode15s | $1.66215 \times 10^{-6}$ | - |
| | ode23s | $2.62933 \times 10^{-6}$ | - |

**Table 4.** Numerical results for test problem 3.

| h | Method | Maxe | Time |
|---|---|---|---|
| $10^{-2}$ | SDIBBDF | $6.17982 \times 10^{-1}$ | $7.86625 \times 10^{-6}$ |
| | DIBBDF | $4.67257 \times 10^{0}$ | $2.78780 \times 10^{-4}$ |
| | BBDF | $1.21585 \times 10^{-2}$ | $4.03164 \times 10^{-4}$ |
| | ode15s | $3.94538 \times 10^{-3}$ | - |
| | ode23s | $1.56018 \times 10^{-3}$ | - |
| $10^{-4}$ | SDIBBDF | $8.04397 \times 10^{-5}$ | $1.16277 \times 10^{-5}$ |
| | DIBBDF | $4.76597 \times 10^{-3}$ | $1.31756 \times 10^{-3}$ |
| | BBDF | $4.78817 \times 10^{-3}$ | $4.48201 \times 10^{-3}$ |
| | ode15s | $5.94372 \times 10^{-5}$ | - |
| | ode23s | $8.46852 \times 10^{-5}$ | - |
| $10^{-6}$ | SDIBBDF | $8.32566 \times 10^{-9}$ | $5.79741 \times 10^{-3}$ |
| | DIBBDF | $4.90299 \times 10^{-5}$ | $6.79440 \times 10^{-1}$ |
| | BBDF | $4.90323 \times 10^{-5}$ | $8.49183 \times 10^{-1}$ |
| | ode15s | $2.63297 \times 10^{-6}$ | - |
| | ode23s | $1.09101 \times 10^{-5}$ | - |

**Table 4.** *Cont.*

| h | Method | Maxe | Time |
|---|---|---|---|
| $10^{-8}$ | SDIBBDF | $3.79303 \times 10^{-9}$ | $3.60277 \times 10^{-1}$ |
| | DIBBDF | $4.90438 \times 10^{-7}$ | $7.80285 \times 10^{0}$ |
| | BBDF | $4.90444 \times 10^{-7}$ | $9.73183 \times 10^{0}$ |
| | ode15s | $2.63297 \times 10^{-6}$ | - |
| | ode23s | $1.09756 \times 10^{-5}$ | - |

**Table 5.** Numerical results for test problem 4.

| h | Method | Maxe | Time |
|---|---|---|---|
| $10^{-2}$ | SDIBBDF | $1.29000 \times 10^{2}$ | $9.18958 \times 10^{-6}$ |
| | DIBBDF | $9.88705 \times 10^{19}$ | $5.24785 \times 10^{-4}$ |
| | BBDF | $5.96778 \times 10^{20}$ | $7.35594 \times 10^{-4}$ |
| | ode15s | $8.40412 \times 10^{-3}$ | - |
| | ode23s | $5.79277 \times 10^{-3}$ | - |
| $10^{-4}$ | SDIBBDF | $1.10568 \times 10^{-2}$ | $9.96719 \times 10^{-4}$ |
| | DIBBDF | $5.59762 \times 10^{-2}$ | $6.17434 \times 10^{-2}$ |
| | BBDF | $5.67153 \times 10^{-2}$ | $9.12839 \times 10^{-2}$ |
| | ode15s | $1.66867 \times 10^{-4}$ | - |
| | ode23s | $2.69421 \times 10^{-4}$ | - |
| $10^{-6}$ | SDIBBDF | $1.24240 \times 10^{-6}$ | $4.50796 \times 10^{-2}$ |
| | DIBBDF | $7.33652 \times 10^{-4}$ | $4.38599 \times 10^{0}$ |
| | BBDF | $7.34010 \times 10^{-4}$ | $6.95911 \times 10^{0}$ |
| | ode15s | $2.74512 \times 10^{-6}$ | - |
| | ode23s | $1.54152 \times 10^{-6}$ | - |
| $10^{-8}$ | SDIBBDF | $5.98807 \times 10^{-9}$ | $5.45829 \times 10^{-1}$ |
| | DIBBDF | $7.35736 \times 10^{-6}$ | $1.36542 \times 10^{1}$ |
| | BBDF | $7.35740 \times 10^{-6}$ | $3.23160 \times 10^{1}$ |
| | ode15s | $2.74508 \times 10^{-6}$ | - |
| | ode23s | $1.48899 \times 10^{-5}$ | - |

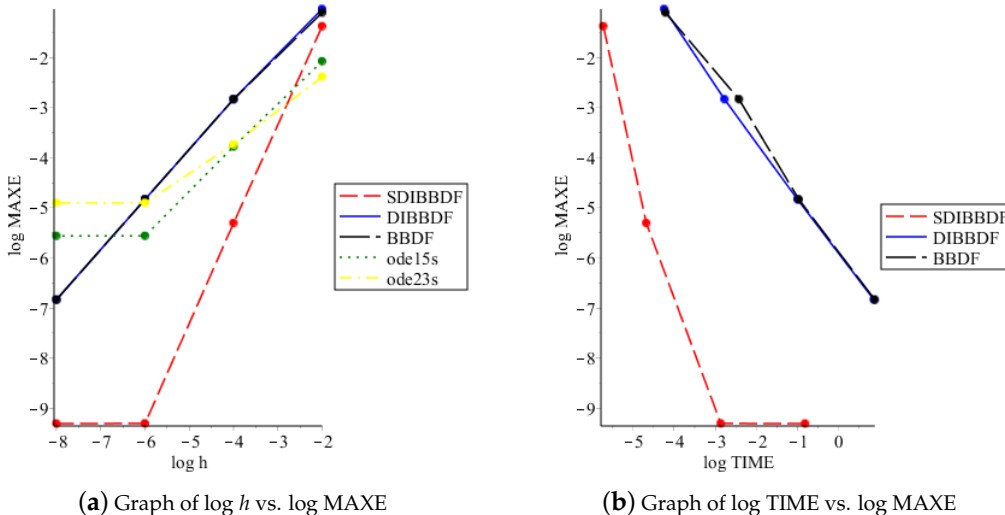

(**a**) Graph of log *h* vs. log MAXE　　　　　　(**b**) Graph of log TIME vs. log MAXE

**Figure 5.** Efficiency curves for test problem 1.

**Table 6.** Numerical results for test problem 5.

| h | Method | Maxe | Time |
|---|---|---|---|
| $10^{-2}$ | SDIBBDF | $3.58622 \times 10^{-1}$ | $2.97236 \times 10^{-6}$ |
| | DIBBDF | $8.79522 \times 10^{24}$ | $3.21313 \times 10^{-4}$ |
| | BBDF | $1.14580 \times 10^{25}$ | $5.30833 \times 10^{-4}$ |
| | ode15s | $2.06838 \times 10^{-1}$ | - |
| | ode23s | $2.06181 \times 10^{-1}$ | - |
| $10^{-4}$ | SDIBBDF | $3.99569 \times 10^{-5}$ | $7.78610 \times 10^{-5}$ |
| | DIBBDF | $8.15054 \times 10^{-3}$ | $2.52951 \times 10^{-3}$ |
| | BBDF | $8.16801 \times 10^{-3}$ | $5.78401 \times 10^{-3}$ |
| | ode15s | $2.06980 \times 10^{-1}$ | - |
| | ode23s | $2.06991 \times 10^{-1}$ | - |
| $10^{-6}$ | SDIBBDF | $3.99999 \times 10^{-9}$ | $4.71637 \times 10^{-2}$ |
| | DIBBDF | $8.22463 \times 10^{-5}$ | $1.30103 \times 10^{-1}$ |
| | BBDF | $8.22481 \times 10^{-5}$ | $3.71753 \times 10^{-1}$ |
| | ode15s | $2.06857 \times 10^{-1}$ | - |
| | ode23s | $2.07027 \times 10^{-1}$ | - |
| $10^{-8}$ | SDIBBDF | $7.53686 \times 10^{-10}$ | $4.13693 \times 10^{-1}$ |
| | DIBBDF | $8.22532 \times 10^{-7}$ | $6.73194 \times 10^{1}$ |
| | BBDF | $8.22546 \times 10^{-7}$ | $9.92184 \times 10^{1}$ |
| | ode15s | $2.06873 \times 10^{-1}$ | - |
| | ode23s | $2.07004 \times 10^{-1}$ | - |

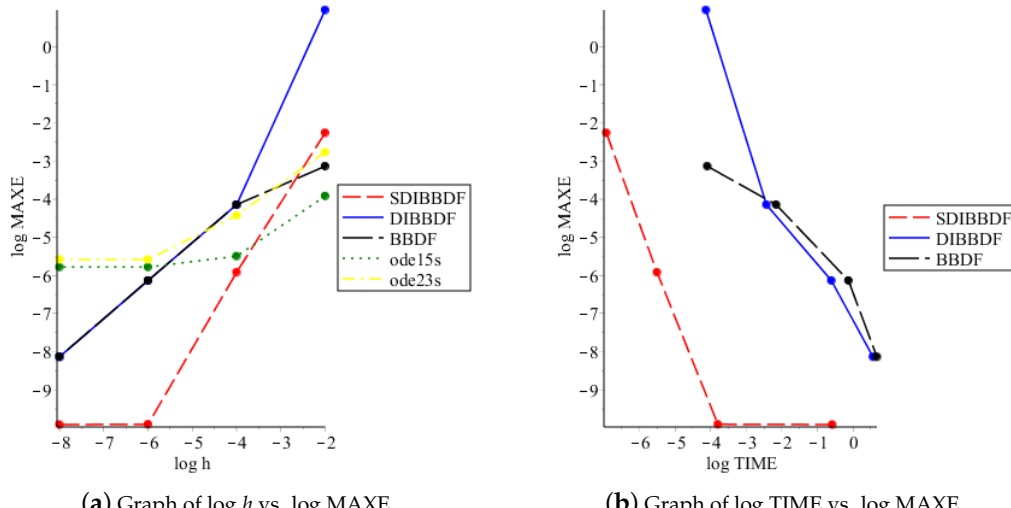

(**a**) Graph of log *h* vs. log MAXE      (**b**) Graph of log TIME vs. log MAXE

**Figure 6.** Efficiency curves for test problem 2.

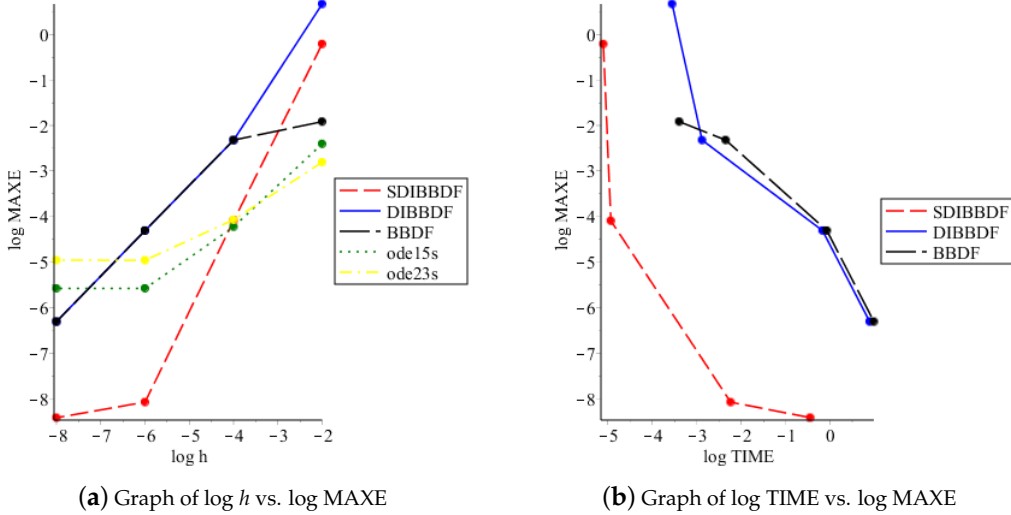

(**a**) Graph of log *h* vs. log MAXE　　　　　(**b**) Graph of log TIME vs. log MAXE

**Figure 7.** Efficiency curves for test problem 3.

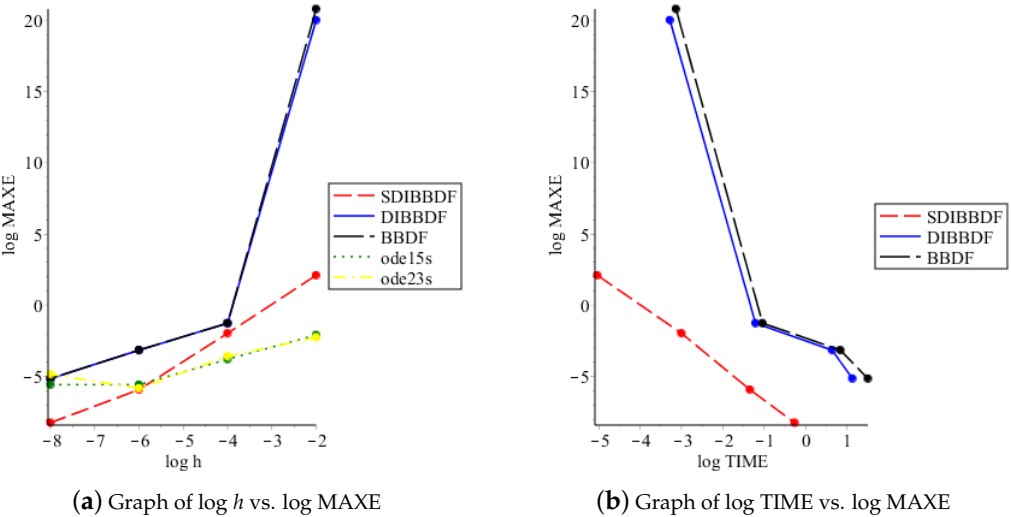

(**a**) Graph of log *h* vs. log MAXE　　　　　(**b**) Graph of log TIME vs. log MAXE

**Figure 8.** Efficiency curves for test problem 4.

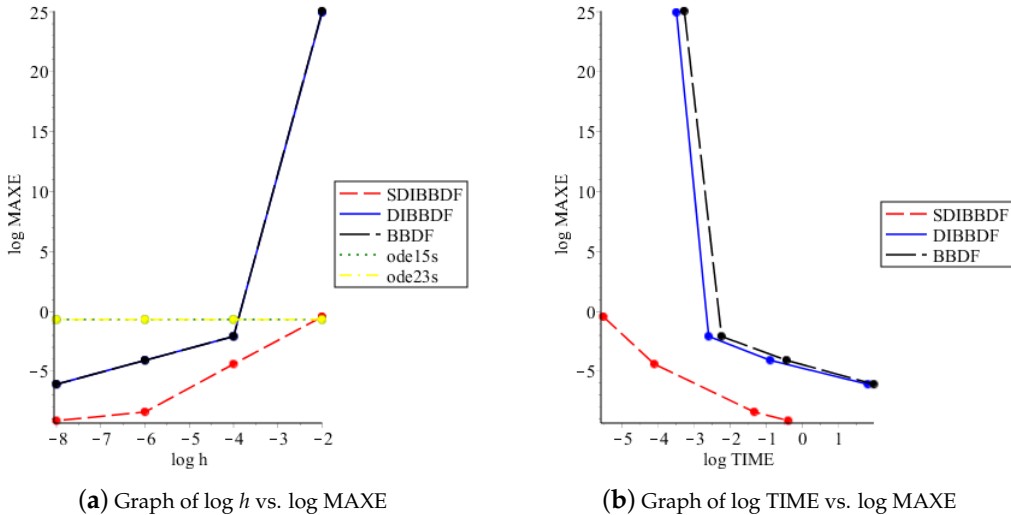

(**a**) Graph of log *h* vs. log MAXE　　　　　(**b**) Graph of log TIME vs. log MAXE

**Figure 9.** Efficiency curves for test problem 5.

Matlab solvers, ode15s and ode23s, are chosen as comparing methods since both solvers are designed for solving stiff problems. Ode15s is a variable order solver based on the numerical differentiation formulas. For certain occasions, it uses the BDFs method. On the other hand, ode23s is a one-step solver based on a modified Rosenbrock formula of order 2. Since results for SDIBBDF, DIBBDF and FIBBDF methods are computed by using C++ programming languages, the TIME of those methods cannot be compared with the one from Matlab.

From the results, one can observed that Matlab solvers, ode15s and ode23s, managed to obtain smaller MAXE than the SDIBBDF method when computing solutions for $h = 10^{-2}$ for every test problems. Numerical results in Table 3 show that SDIBBDF method produced slightly bigger MAXE than the fully implicit method when $h = 10^{-2}$. This is due to the step size restriction as discussed earlier in Section 2.3. Since test problem 2 has $\lambda = -100$ hence, it requires $h < 0.006240609349$.

Not only that, but step size restriction is also experienced when $h = 10^{-2}$ is used to solve test problem 3 with $\lambda = -100$ and $-1$. Notice that the SDIBBDF, DIBBDF and BBDF method produced bigger MAXE for test problem 4 and 5 than the other test problems for stepsize $h = 0.01$. The errors obtained by both test problems are affected by the nature of the system where test problem 4 involved a highly stiff ODEs with $\lambda = -1000$ and $-1$ while test problem 5 consists of complex eigenvalues, $\lambda = -40 - 40i, -40 + 40i$ and $-2$.

On the contrary, efficiency of the SDIBBDF method is proven when smaller stepsizes are used. It is capable of outperforming the other comparing methods for $h = 10^{-4}$, $h = 10^{-6}$ and $h = 10^{-8}$ for each problems of stiff ODEs tested.

As for TIME, the proposed method is capable of executing solutions faster than the fully and diagonally implicit methods for each test problem. This is motivated by the implementation of singly diagonally implicit properties that had proved to improve efficiency of the method. Figures 5b–9b show that smaller step sizes require longer computational time due to larger number of steps involved.

## 4. Conclusions

The 2-point SDIBBDF method of order 2 for solving stiff ODEs is successfully derived by implementing the SDIRK properties of one-step method to the multistep block method. Stability analysis of the proposed method shows that it is zero-stable and with the *A*-stable characteristics which makes it fit for solving stiff problems.

Based on Section 2.3, we analysed the relation between step size and eigenvalue of stiff ODEs which gives effect on the MAXE produced. Hence, we will be able to set the best step size to used for solving stiff ODEs based on its eigenvalues in order to ensure good performance of SDIBBDF method to approximate solutions of the problems.

From numerical results, it is proved that the SDIBDDF method produced better accuracy for smaller stepsizes than the other comparing methods, and capable to execute solutions faster than the DIBBDF and fully implicit BBDF method. Effects of the step size restriction are also shown on the numerical results and analysed further.

Therefore, we can conclude that the SDIBBDF method can be used as an alternative solver for solving stiff ODEs.

**Author Contributions:** Conceptualization, S.J.A. and I.S.M.Z.; Data curation, S.J.A. and Z.B.I.; Formal analysis, S.J.A.; Methodology, S.J.A.; Supervision, Z.B.I.; Validation, Z.B.I. and I.S.M.Z.; Visualization, S.J.A.; Writing—original draft preparation, S.J.A.; Writing—review and editing, S.J.A. and Z.B.I.

**Funding:** This research was funded by the Universiti Putra Malaysia under the Putra-IPS Grant (project no.: GPS-IPS/2017/9518800).

**Conflicts of Interest:** The authors declare no conflict of interest.

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
