# Peer review of "Stability Analysis of Singly Diagonally Implicit Block Backward Differentiation Formulas for Stiff Ordinary Differential Equations"

_mathematics, doi:10.3390/math7020211_

Round 1

Reviewer 1 Report

The paper presents a work with a brilliant idea how to improve the performance of linear multistep methods for stiff problems in a manner it is made for RK methods, and the description of mathematical aspects is very detailed and easy to understand. But it is a great broblem of many studies in numerical analysis including the current paper that the experimental part of the work does not involve any well established and widely used methods. For the current study, the classical BDF is the one, and many questions about practical applicability of the SDIBBDF arise. For instance, the stability region of the proposed method seems to be almost similar to the stability region of the BDF2 method, so no profit can be seen (see diagrams in Süli, E., & Mayers, D. F. An Introduction to Numerical Analysis, Cambridge University Press, 2003, p. 350).

Also, since SDIBBDF outperforms the other two methods so well, its comparison to BDF or even RK is urgent. In my experience, this can be an evidence of some "optimal" method coefficient design (when features of the other compared methods like stability regions are the same). Otherwise, only some combined approaches can give such superiority on simple test problems within methods of the same class (like semi-implicit methods or adaptive switching solvers, see e.g. Karimov, A. I., et al "Adaptive explicit-implicit switching solver for stiff ODEs." Young Researchers in Electrical and Electronic Engineering (EIConRus), 2017 IEEE Conference of Russian. IEEE, 2017).

Notice, that namely the BDF methods (not the BBDF or other block modifications) are used in software packages like LabVIEW, Multisim and other SPICE family design programs (their stability is considered in many papers, see e.g. Gubian, P., & Zanella, M. (1991, June). Stability properties of integration methods in SPICE transient analysis. In Circuits and Systems, 1991., IEEE International Sympoisum on (pp. 2701-2704). IEEE.). In fact, the test problems authors evaluate may be too simple to benefit from "diagonal implicitness", but more high-dimensional ODEs may reveal the difference. If is does, the paper will comprise a notable contribution to the progress of numerical integration methods.

Author Response

You may find response to the reviewer's comments as attached

Reviewer 2 Report

The paper presents a combination of two methods described in resources cited in references.

However, I have two comments to what and how is presented.

1) Approximately 1/3 of the references, precisely 10 of 28 cited resources, are written by authors who come from the same university. Moreover, 6 of the 10 works are Ph.D. or Master theses. This is quite unusual. What is even more unusual and the main issue of the paper, is that comparison of errors and computation times are done only against results coming from the same university. This fact significantly decreases value of the paper. It is necessary to compare values against results obtained by researchers in other parts of the world. Nevertheless, I have to admit that References are done in such a smart way that it is not easy to find out what and why is the problem. The 10 problematic references are highlighted in the attached PDF file.

2) I added several comments to the attached PDF file. Most of them are related to using symbols without giving them a meaning in the text before. It is necessary to go through the whole paper and check the consistency of symbols and definitions used. Without that, it is difficult to read and understand the paper.

Also, English language should be checked, especially with respect to mathematical formulations.

Author Response

You may find response to the reviewer's comments as attached.

Round 2

Reviewer 1 Report

Dear authors,

thank you for good work on reconsidering the manuscript. The paper is now more clear and is ready for publication. 

Nevertheless, I would like to notice that in my humble opinion, the reply that the comparison with non-block and one-step methods is irrelevant, may be controversial. The main aim of any improvement in numerical methods is to design the fastest  and the most accurate algorithm for practical application regardless what kind of methods it belongs to. So, I expected to see some elaborate performance tests on really high-dimensional problems (e.g. any practical circuit simulation). It was not done, but I expect to see that in the future work and hope your method will find a wide practical application.

 Also, I agree that the basic principles of the research on methods have not been changed since 1960s-1970s. But I would recommend to add more actual references in the future publications, because citing modern authors will increase the visibility of your work to the scientific society.

Anyway, I appreciate your notable work and recommend the paper for publication.

Author Response

Greetings,

Thank you for your comments and suggestions, we really appreciate them, and thank you for recommended our paper for publication. Please find the response to reviewer comments in attached file.

Thank you.

Reviewer 2 Report

Thank you for the revision and improvements done. However,

1) the main issue of the paper still is that comparison of errors and computation times are done only against results coming from the same university. This fact significantly decreases value of the paper. If you want to increase the value of your manuscript, it is necessary to compare values against results obtained by researchers in other parts of the world. I understand that comparison of computation time is important from your point of view, but for most readers, accuracy is the first. It is necessary to introduce at least 2 examples where the comparison of accuracy will be done against results published in peer-reviewed journals or monographs. Without that, significance of content of your manuscript remains low.

2) I added several comments to the attached PDF file. It is necessary to go through the whole paper and check the correctness of formulas and consistency of symbols and definitions used. Without that, it is difficult to read and understand the paper.

Author Response

Greetings,

Thank you for your comments and suggestions. Please find the response to reviewer comments in attached file.

Thank you.

Round 3

Reviewer 2 Report

Dear authors, thank you for submitting revised version. You have done a good job.

However, I still have a few comments. Please find them in the PDF file attached.

I am concerned about expressions (11) and (12) in this version (v3). As soon as we clarify the issue, I am willing to recommend the paper for publication.

Author Response

(The authors gave the same response as above.)

Round 4

Reviewer 2 Report

Dear Authors, thank you for the explanation. It sounds OK to me.

Just one more suggestion, on page 4, line 93, I suggest to use "and" instead of "with".